# On Breiman's Dilemma in Neural Networks: Success and Failure of Normalized Margins

## Abstract

A belief persists long in machine learning that enlargement of margins over training data accounts for the resistance of models to overfitting by increasing the robustness. Yet Breiman shows a dilemma (Breiman, 1999) that a uniform improvement on margin distribution *does not* necessarily reduces generalization error. In this paper, we revisit Breiman's dilemma in deep neural networks with recently proposed normalized margins using Lipschitz constant bound by spectral norm products. With both simplified theory and extensive experiments, Breiman's dilemma is shown to rely on dynamics of normalized margin distributions, that reflects the trade-off between model expression power and data complexity. When the complexity of data is comparable to the model expression power in the sense that training and test data share similar phase transitions in normalized margin dynamics, Rademacher complexities of Lipschitz-normalized networks can be regarded as small constants and two efficient ways are derived via classic margin-based generalization bounds to successfully predict the trend of generalization error. On the other hand, over-expressed models that exhibit uniform improvements on training normalized margins may lose such a prediction power and fail to prevent the overfitting.

## 1 Introduction

Margin, as a measurement of the robustness allowing some perturbations on classifier without changing its decision on training data, has a long history in characterizing the performance of classification algorithms in machine learning. As early as Novikoff (1962), it played a central role in the proof on finite-stopping or convergence of perceptron algorithm when training data is separable. Equipped with convex optimization technique, a plethora of large margin classifiers are triggered by support vector machines (Cortes & Vapnik, 1995; Vapnik, 1998). AdaBoost, an iterative algorithm to combine an ensemble of classifiers proposed by Freund & Schapire (1997), often exhibits a resistance to overfitting phenomenon that during the training process the generalization error keeps on non-increasing when the training error drops to zero. Toward deciphering the such a resistance of overfitting phenomenon, Schapire et al. (1998) proposed an explanation that the training process keeps on improving a notion of classification margins in boosting, among later works on consistency of boosting with early stopping regularization (Bühlmann & Yu, 2002; Zhang & Yu, 2005; Yao et al., 2007). Lately such a resistance to overfitting is again observed in deep neural networks with overparameterized models (Zhang et al., 2016). A renaissance of margin theory is proposed by Bartlett et al. (2017) with a normalization of network using Lipschitz constants bounded by products of operator spectral norms. It inspires many further investigations in various settings (Miyato et al., 2018; Neyshabur et al., 2018; Liao et al., 2018).

However, the improvement of margin distributions does not necessarily guarantee a better generalization performance, which is at least traced back to (Breiman, 1999) in his effort to understanding AdaBoost. In this work, Breiman designed an algorithm *arc-gv* such that the margin can be maximized via a prediction game, then he demonstrated an example that one can achieve uniformly larger margin distributions on training data than AdaBoost but suffer a higher generalization error. In the end of this paper, Breiman made the following comments with a dilemma:

*"The results above leave us in a quandary. The laboratory results for various arcing algorithms are excellent, but the theory is in disarray. The evidence is that if we try too hard to make the margins*

*larger, then overfitting sets in. ... My sense of it is that we just do not understand enough about what is going on."*

Breiman's dilemma triggers some further explorations to understand the limitation of margin theory in boosting (Reyzin & Schapire, 2006; Wang et al., 2008; 2011). In particular, Reyzin & Schapire (2006) points out that the trees found by *arg-gv* have larger model complexity in terms of deeper average depth than AdaBoost, suggesting that margin maximization in *arc-gv* does not necessarily control the model complexity. The latter works provide tighter bounds based on VC-dimension and optimized quantile training margins, which however do not apply to over-parametrized models in deep neural networks and the case where the training margin distributions are uniformly improved.

In this paper, we are going to revisit Breiman's dilemma in the scenario of deep neural networks. Both the success and failure can be seen on normalized margin based bounds on generalization error. First of all, let's look at the following illustration example.

**Example** (Breiman's Dilemma with a CNN). *A basic 5-layer convolutional neural network of $c$ channels (see Section 3 for details) is trained with CIFAR-10 dataset whose 10 percent labels are randomly permuted. When $c = 50$ with $92,610$ parameters, Figure 1 (a) shows the training error and generalization (test) error in solid curves. From the generalization error in (a) one can see that overfitting indeed happens after about 10 epochs, despite that training error continuously drops down to zero. One can successfully predict such an overfitting phenomenon from Figure 1 (b), the evolution of normalized margin distributions defined later in this paper. In (b), while small margins are monotonically improved during training, large margins undergoes a phase transition from increase to decrease around 10 epochs such that one can predict the tendency of generalization error in (a) using large margin dynamics. Two particular sections of large margin dynamics are highlighted in (b), one at 8.3 on $x$-axis that measures the percentage of normalized training margins no more than 8.3 (training margin error) and the other at 0.8 on $y$-axis that measures the normalized margins at quantile $q = 0.8$ (i.e. $1/\hat{\gamma}_{q,t}$). Both of them meet the tendency of generalization error in (a) and find good early stopping time to avoid overfitting. However, as we increase the channel number to $c = 400$ with about $5.8M$ parameters and retrain the model, (c) shows a similar overfitting phenomenon in generalization error; on the other hand, (d) exhibits a monotonic improvement of normalized margin distributions without a phase transition during the training and thus fails to capture the overfitting. This demonstrates the Breiman's dilemma in CNN.*

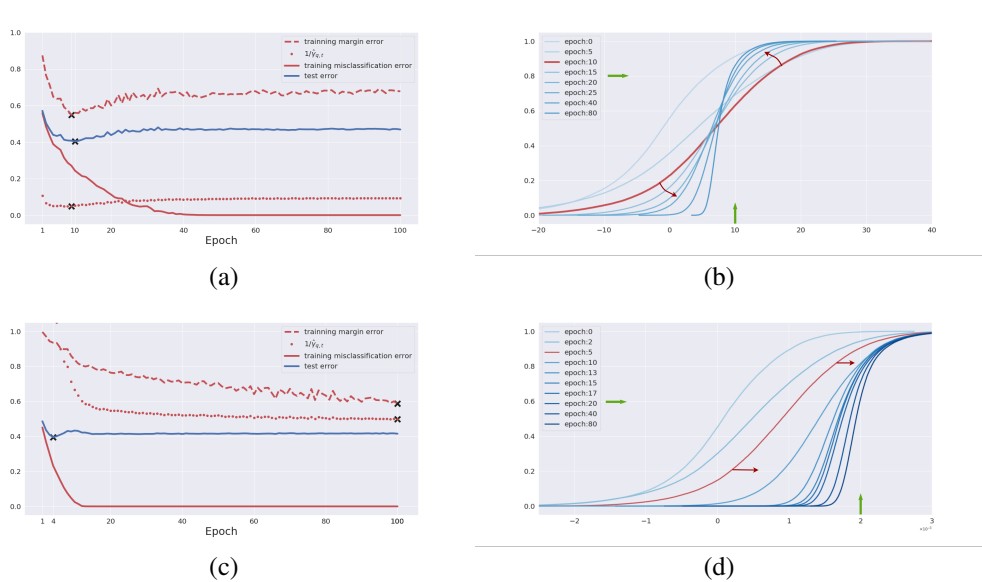

Figure 1: Demonstration of Breiman's Dilemma in Convolutional Neural Networks.

A key insight behind this dilemma, is that one needs a trade-off between the model expression power and the complexity of the dataset to endorse margin bounds a prediction power. On one hand, when the model has a limited expression power relative to the training dataset, in the sense that the

training margin distributions CAN NOT be uniformly improved during training, the generalization or test error may be predicted from dynamics of normalized margin distributions. On the other hand, if we push too hard to improve the margin by giving model too much degree of freedom such that the training margins are uniformly improved during training process, the predictability may be lost. A trade-off is thus necessary to balance the complexity of model and dataset, otherwise one is doomed to meet Breiman's dilemma when the models arbitrarily increase the expression power.

The example above shows that the expression power of models relative to the complexity of dataset, can be observed from the dynamics of normalized margins in training, instead of counting the number of parameters in neural networks. In the sequel, our main contributions are to make these precise by revisiting the Rademacher complexity bounds with Lipschitz constants (Bartlett et al., 2017).

- With the Lipschitz-normalized margins, a linear inequality is established between training margin and test margin in Theorem 1. When both training and test normalized margin distributions undergo similar phase transitions on increase-decrease during the training process, one may predict the generalization error based on the training margins as illustrated in Figure 1.

- In a dual direction, one can define a *quantile margin* via the inverse of margin distribution functions, to establish another linear inequality between the inverse quantile margins and the test margins as shown in Theorem 2. Quantile margin is far easier to tune in practice and enjoys a stronger prediction power exploiting an adaptive selection of margins along model training.

- In all cases, Breiman's dilemma may fail both of the methods above when dynamics of normalized training margins undergo different phase transitions to that of test margins during training, where a uniform improvement of margins results in overfitting.

Section 2 describes our method to derive the two linear inequalities of generalization bounds above. Extensive experimental results are shown in Section 3 and Appendix with basic CNNs, AlexNet, VGG, ResNet, and various datasets including CIFAR10, CIFAR100, and mini-Imagenet.

## 2 METHOD

Let $\mathcal{X}$ be the input space (e.g. $\mathcal{X} \subset \mathbb{R}^{C \times W \times H}$ in image classification) and $\mathcal{Y} := \{1, \ldots, K\}$ be the space of $K$ classes. Consider a sample set of $n$ observations $S = \{(x_1, y_1), \ldots, (x_n, y_n) : x_i \in \mathcal{X}, y_i \in \mathcal{Y}\}$ that are drawn i.i.d. from $P_{X,Y}$. For any function $f : \mathcal{X} \to \mathbb{R}$, let $\mathbb{P}f = \int_{\mathcal{X}} f(X) dP$ be the population expectation and $\mathbb{P}_n f = (1/n) \sum_{i=1}^{n} f(x_i)$ be the sample average.

Define $\mathcal{F}$ to be the space of functions represented by neural networks,

$$\mathcal{F} = \{f : \mathcal{X} \to \mathbb{R}^K, f(x) = W_l \sigma_l(x_l) + b_l, x_i = \sigma_i(W_{i-1}x_{i-1} + b_{i-1}), i = 1, \ldots, l, x_0 = x\}, \quad (1)$$

where $l$ is the depth of the network, $W_i$ is the weight matrix corresponding to a linear operator on $x_i$ and $\sigma_i$ stands for either element-wise activation function (e.g. ReLU) or pooling operator that are assumed to be Lipschitz bounded with constant $L_{\sigma_i}$ and satisfying $\sigma_i(0) = 0$. For example, in convolutional network, $W_i x_i + b_i = w_i * x_i + b_i$ where $*$ stands for the convolution between input tensor $x_l$ and kernel tensor $w_l$. We equip $\mathcal{F}$ with the Lipschitz semi-norm, for each $f$,

$$\|f\|_{\mathcal{F}} := \sup_{x \neq x'} \frac{\|f(x) - f(x')\|_2}{\|x - x'\|_2} \leq L_\sigma \prod_{i=1}^{l} \|W_i\|_\sigma := L_f, \quad (2)$$

where $\| \cdot \|_\sigma$ is the spectral norm and $L_\sigma = \prod_{i=1}^{L} L_{\sigma_i}$. For all the examples in this paper, we use ReLU activation $\sigma_i$ that leads to $L_{\sigma_i} = 1$. Moreover we consider the following family of hypothesis mapping,

$$\mathcal{H} = \{h(x) = [f(x)]_y : \mathcal{X} \to \mathbb{R}, f \in \mathcal{F}, y \in \mathcal{Y}\}, \quad (3)$$

where $[\cdot]_j$ denotes the $j^{\text{th}}$ coordinate and we further define the following class induced by Lipschitz semi-norm bound on $\mathcal{F}$,

$$\mathcal{H}_L = \{h(x) = [f(x)]_y : \mathcal{X} \to \mathbb{R}, h(x) = [f(x)]_y \in \mathcal{H} \text{ with } \|f\|_{\mathcal{F}} \leq L, y \in \mathcal{Y}\}. \quad (4)$$

Lastly, rather than merely looking at whether a prediction $f(x)$ on $y$ is correct or not, we also consider the margin defined as $\zeta(f(x), y) = [f(x)]_y - \max_{\{j:j \neq y\}} [f(x)]_j$. Therefore, we can define the *ramp loss* and *margin error* depending on the confidence of predictions. Given two thresholds $\gamma_2 > \gamma_1 \geq 0$, define a ramp loss to be

$$\ell_{(\gamma_1, \gamma_2)}(\zeta) = \begin{cases} 1 & \zeta < \gamma_1, \\ -\frac{1}{\Delta}(\zeta - \gamma_2) & \gamma_1 \leq \zeta \leq \gamma_2, \\ 0 & \zeta > \gamma_2, \end{cases}$$

where $\Delta := \gamma_2 - \gamma_1$. In particular $\gamma_1 = 0$ and $\gamma_2 = \gamma$, we also write $\ell_\gamma = \ell_\gamma$ for simplicity. Define the margin error to measure if $f$ has margin no more than a threshold $\gamma$,

$$e_\gamma(f(x), y) = \begin{cases} 1 & \zeta(f(x), y) \leq \gamma \\ 0 & \zeta(f(x), y) > \gamma \end{cases}. \tag{5}$$

In particular, $e_0(f(x), y)$ is the common mis-classification error and $\mathbb{E}[e_0(f(x), y)] = \mathbb{P}[\zeta(f(x), y) < 0]$. Note that $e_0 \leq \ell_\gamma \leq e_\gamma$, and $\ell_\gamma$ is Lipschitz bounded by $1/\gamma$.

The central question we try to answer is, *can we find a proper upper bound to predict the tendency of the generalization error along training, such that one can early stop the training near the epoch that $\mathbb{P}[\zeta(f_t(x), y) < 0]$ is minimized?* The answer is both a *yes* and a *no!*

We begin with the following lemma, as a typical result in multi-label classification from the uniform law of large numbers (Koltchinskii et al., 2002).

**Lemma 2.1.** *Given a $\gamma_0 > 0$, then, for any $\delta \in (0, 1)$, with probability at least $1 - \delta$, the following holds for any $f \in \mathcal{F}$ with $\|f\|_{\mathcal{F}} \leq L$,*

$$\mathbb{E}[\ell_{\gamma_0}(f(x), y)] \leq \frac{1}{n} \sum_{i=1}^n [\ell_{\gamma_0}(f(x_i), y_i)] + \frac{2K^2}{\gamma_0} \mathcal{R}_n(\mathcal{H}_L) + \sqrt{\frac{\log(1/\delta)}{2n}} \tag{6}$$

*where*

$$\mathcal{R}_n(\mathcal{H}_L) = \mathbb{E}_{x_i, \varepsilon_i} \sup_{h \in \mathcal{H}_L} \frac{1}{n} \sum_{i=1}^n \varepsilon_i h(x_i) \tag{7}$$

*is the Rademacher complexity of function class $\mathcal{H}_L$ with respect to $n$ samples, and the expectation is taken over $x_i, \varepsilon_i$, $i = 1, ..., n$.*

Unfortunately, direct application of such bound for a constant $\gamma_0$ will suffer from the so-called *scaling problem*. The following proposition gives an lower bound of Rademacher complexity term, whose proof is provided in Appendix D.

**Proposition 1.** *Consider the networks with ReLU activation functions. For any $L > 0$, there holds,*

$$\mathcal{R}_n(\mathcal{H}_L) \geq CL\mathbb{E}_S[\sqrt{x_1^2 + \ldots + x_n^2}] \tag{8}$$

*where $C > 0$ is a constant that does not depend on $S$.*

The lemma tells us if $L \to \infty$, upper bound (6) becomes trivial since $\mathcal{R}_n(\mathcal{H}_L) \to \infty$. In fact, both Telgarsky (2013) and Soudry et al. (2018) show that with gradient descent, the norm of estimator's weight in logistic regression and general boosting (including exponential loss), respectively, will go to infinity at a growth rate $\log(t)$ when the data is linearly separable. As for the deep neural network with cross-entropy loss, the input of last layer is usually be viewed as features extracted from original input. Training the last layer with other layers fixed is exactly a logistic regression, and the feature is linearly separable as long as the training error achieves zero. Therefore, without any normalization, the hypothesis space along training has no upper bound on $L$ and the upper bound (6) is useless. Besides, even for a fixed $L$, the complexity term $\mathcal{R}_n(\mathcal{H}_L)$ is computationally intractable.

The first remedy is to restrict our attention on $\mathcal{H}_1$ by normalizing $f$ with its Lipschitz semi-norm $\|f\|_{\mathcal{F}}$ or its upper bounds. Note that a normalized network $\tilde{f} = f/C$ has the same mis-classification error as $f$ for all $C > 0$. For the choice of $C$, it's hard in practice to directly compute the Lipschitz semi-norm of a network, but instead some approximate estimates on the upper bound $L_f$ in (2) are available as discussed in Appendix A. In the sequel, let $\tilde{f} = f/L_f$ be the normalized network and

$\tilde{h} = h/L_f = \zeta(f,y)/L_f = \zeta(\tilde{f},y) \in \mathcal{H}_1$ be the corresponding normalized hypothesis function. Now a simple idea is to regard $\mathcal{R}_n(\mathcal{H}_1)$ as a constant and predict the tendency of generalization error via training margin error of the normalized network, that avoids the scaling problem and the computation of complexity term. The following theorem makes this precise.

**Theorem 1.** *Given $\gamma_1$ and $\gamma_2$ such that $\gamma_2 > \gamma_1 \geq 0$ and $\Delta := \gamma_2 - \gamma_1 \geq 0$, for any $\delta > 0$, with probability at least $1 - \delta$, along the training epoch $t = 1, \ldots, T$, the following holds for each $f_t$,*

$$\mathbb{P}[\zeta(\tilde{f}_t(x), y) < \gamma_1] \leq \mathbb{P}_n 1[\zeta(\tilde{f}_t(x), y) < \gamma_2] + \frac{C_{\mathcal{H}}}{\Delta} + \sqrt{\frac{\log(1/\delta)}{2n}} \tag{9}$$

*where $C_{\mathcal{H}} = 2K^2 \mathcal{R}_n(\mathcal{H}_1)$.*

**Remark.** *In particular, when we take $\gamma_1 = 0$ and $\gamma_2 = \gamma > 0$, the bound above becomes,*

$$\mathbb{P}[\zeta(f_t(x), y) < 0] \leq \mathbb{P}_n[\zeta(\tilde{f}_t(x_i), y_i) < \gamma] + \frac{C_{\mathcal{H}}}{\gamma} + \sqrt{\frac{\log(1/\delta)}{2n}} \tag{10}$$

Theorem 1 says, we can bound the normalized test margin distribution $\mathbb{P}[\zeta(\tilde{f}_t(x), y) < \gamma_1]$ by the normalized training margin distribution $\mathbb{P}_n[\zeta(\tilde{f}_t(x), y) < \gamma_2]$. Recently Liao et al. (2018) investigates for normalized networks, the strong linear relationship between cross-entropy training loss and test loss when the training epochs are large enough. As a contrast, we consider the whole training process and normalized margins. In particular, we hope to predict the trend of generalization error by choosing $\gamma_1 = 0$ and a proper $\gamma$. For this purpose, the following facts are important. First, we do not expect the bound, for example (10), is tight for every choice of $\gamma > 0$, instead we hope there exists some $\gamma$ such that the training margin error nearly monotonically changes with generalization error. Figure 2 shows the existence of such $\gamma$ such that the training margin error successfully recover the tendency of generalization error on CIFAR10 dataset. Moreover, in Appendix Figure 8 shows the rank correlation between training margin error at various $\gamma$ and training/test error. Second, the normalizing factor is not necessarily to be an upper bound of Lipschitz semi-norm. The key point is to prevent the complexity term of the normalized network going to infinity. Since for any constant $c > 0$, normalization by $\bar{L} = cL$ works in practice where the constant could be absorbed to $\gamma$, we could ignore the Lipschitz constant introduced by general activation functions in the middle layers.

However, it is a natural question whether a reasonable $\gamma$ with prediction power exists. A simple example in Figure 1 shows, once the training margin distribution is uniformly improved, dynamic of training margin error fails to detect the minimum of generalization error in the early stage. This is because when network structure becomes complex enough, the training margin distribution could be more easily improved but the the generalization error may overfit. This is exactly the same observation in Breiman (1999) to doubt the margin theory in boosting type algorithms. More detailed discussions will be given in Section 3.2.

The most serious limitation of Theorem 1 lies in we must fix a $\gamma$ along the complete training process. In fact, the first term and second term in the bound (10) vary in the opposite directions with respect to $\gamma$, and thus different $f_t$ may prefer different $\gamma$ for a trade-off. As in Figure 1 (b) of the example, while choosing $\gamma$ is to fix an $x$-coordinate section of margin distributions, its dual is to look for a $y$-section which leads to different margins for different $f_t$. This motivates the *quantile margin* in the following theorem. Let $\hat{\gamma}_{q,f}$ be the $q^{\text{th}}$ *quantile margin* of the network $f$ with respect to sample $S$,

$$\hat{\gamma}_{q,f} = \inf \{\gamma : \mathbb{P}_n 1[\zeta(f(x_i), y_i) \leq \gamma] \geq q\}. \tag{11}$$

**Theorem 2.** *Assume the input space is bounded by $M > 0$, that is $\|x\|_2 \leq M$, $\forall x \in \mathcal{X}$. Given a quantile $q \in [0, 1]$, for any $\delta \in (0, 1)$ and $\tau > 0$, the following holds with probability at least $1 - \delta$ for all $f_t$ satisfying $\hat{\gamma}_{q,\tilde{f}_t} > \tau$,*

$$\mathbb{P}[\zeta(f_t(x), y) < 0] \leq C_q + \frac{C_{\mathcal{H}}}{\hat{\gamma}_{q,\tilde{f}_t}} \tag{12}$$

$C_q = q + \sqrt{\frac{\log(2/\delta)}{2n}} + \sqrt{\frac{\log \log_2(4(M+l)/\tau)}{n}}$ *and* $C_{\mathcal{H}} = 4K^2 \mathcal{R}_n(\mathcal{H}_1)$.

**Remark.** *We simply denote $\gamma_{q,t}$ for $\gamma_{q,\tilde{f}_t}$ when there is no confusion.*

Compared with the bound (10), (12) make the choice of $\gamma$ varying with $f_t$ and the cost is an additional constant term $C_q^2$ and the constraint $\hat{\gamma}_{q,t} > \tau$ that typically holds for large enough $q$ in practice. In applications, stochastic gradient descent (SGD) often effectively improves the training margin distributions along the drops of training errors, a small enough $\tau$ and large enough $q$ usually meet $\hat{\gamma}_{q,t} > \tau$. Moreover, even with the choice $\tau = \exp(-B)$, constant term $\sqrt{[\log\log_2(4(M+l)/\tau)]/n} = O(\sqrt{\log B/n})$ is still negligible and thus very little cost is paid in the upper bound.

In practice, tuning $q \in [0,1]$ is far easier than tuning $\gamma > 0$ directly and setting a large enough $q \geq 0.9$ usually provides us lots of information about the generalization performance. The quantile margin works effectively when the dynamics of large margin distributions reflects the behavior of generalization error, e.g. Figure 1. In this case, after certain epochs of training, the large margins have to be sacrificed to further improve small margins to reduce the training loss, that typically indicates a possible saturation or overfitting in test error.

## 3 EXPERIMENTAL RESULTS

We briefly introduce the network and dataset used in the experiments. For the network, we first consider the convolutional neural network with very simple structure *basic CNN(c)*. The structure is shown in Appendix Figure 7. Basically, it has five convolutional layers with $c$ channels at each and one fully connected layer, where $c$ will be specified in concrete examples. Second, we consider more practical network structure, AlexNet (Krizhevsky et al., 2012), VGGNet-16 (Simonyan & Zisserman, 2014) and ResNet-18 (He et al., 2016). For the dataset, we consider CIFAR10, CIFAR100 (Krizhevsky & Hinton, 2009) and Mini-ImageNet (Vinyals et al., 2016).

The spirit of the following experiments is to show, *when and how, the margin bound could be used to predict the tendency of generalization or test error along the training path?*

### 3.1 SUCCESS: TRAINING MARGIN ERROR AND QUANTILE MARGIN

This section is to apply Theorem 1 and Theorem 2 to predict the tendency of generalization error. Let's firstly consider training a basic CNN(50) on CIFAR10 dataset with and without random noise. The relations between generalization error and *training margin error* $e_\gamma(\tilde{f}(x), y)$ with $\gamma = 9.8$, *inverse quantile margin* $1/\hat{\gamma}_{q,t}$ with $q = 0.6$ are shown in Figure 2. In this simple example where the net is light and the dataset is simple, the linear bounds (9) and (12) show a good prediction power: they stop either near the epoch of sufficient training (Left, original data) or where even an overfitting occurs (Right, 10 percents label corrupted).

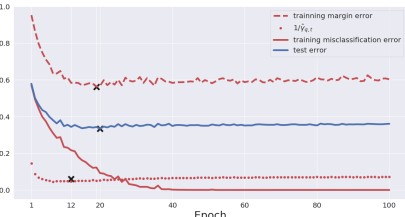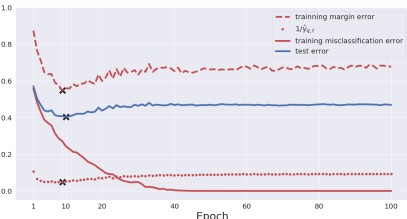

Figure 2: Success examples. Net structure: basic CNN (50). Dataset: Original CIFAR10 (Left) and CIFAR10 with 10 percents label corrupted (Right). In each figure, we show training error (red solid), training margin error $\gamma = 10$ (red dash) and inverse quantile margin (red dotted) with $q = 0.6$ and generalization error (blue solid). The marker "x" in each curve indicates the global minimum along epoch $1, \ldots, T$. Both training margin error and inverse quantile margin successfully predict the tendency of generalization error.

A few discussions are given below.

1. There exists a trade-off on the choice of $\gamma$ from the linear bounds (9) (and parallel arguments hold for $q$). The training margin error with a small $\gamma$ is close to the training error,

while a large $\gamma$ is close to generalization error and it's illustrated in Appendix Figure 8 where we show the *Spearman's $\rho$ rank correlation*[1] between training margin error and training error, generalization error against threshold $\gamma$.

2. The training margin error (or inverse quantile margin) is closely related to the dynamics of training margin distributions. For certain choice of $\gamma$, if the curve of training margin error (with respect to epoch) is V-shape, the corresponding dynamics of training margin distributions will have a *cross-over*, where the low margins have a monotonic increase and the large margins undergo a phase transition from increase to decrease, as illustrated by the red arrow in Figure 1 (b).

3. Dynamics of quantile margins can adaptively select $\gamma_t$ for each $f_t$ without access to the complexity term. Unlike merely looking at the training margin error with a fixed $\gamma$, quantile margin bound (12) shows a stronger prediction power than (10) and even be able to capture more local information as illustrated in Figure 3. The generalization error curve has two valleys corresponding to a local optimum and a global optimum, and the quantile margin curve with $q = 0.95$ successfully identifies both. However, if we consider the dynamics of training margin errors, it's rarely possible to recover the two valleys at the same time since their critical thresholds $\gamma_{t_1}$ and $\gamma_{t_2}$ are different. Another example of ResNet is given in Appendix Figure 9.

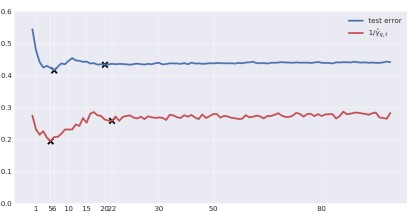 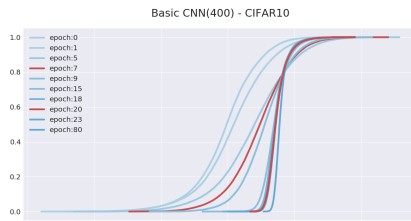

Figure 3: Inverse quantile margin. Net structure: Basic CNN. Dataset: CIFAR10 with 10 percents label corrupted. Left: the dynamic of generalization error (blue) and inverse quantile margin with $q = 0.95$ (red). Two local minimums are marked by "x" in each dynamic. Right: dynamic of training margin distribution and two distributions when local minimum occurs are highlighted with red color. The inverse quantile margin successfully captures two local minimums of test error.

## 3.2 FAILURE: BREIMAN'S DILEMMA IN OVER-PARAMETERIZED MODELS

In this section, we explore the normalized margin dynamics with over-parameterized models whose expression power might be greater than data complexity. We conduct experiments in the following two scenarios.

1. In the first experiment shown in Figure 4, we fix the dataset to be CIFAR10 with 10 percent of labels randomly permuted, and gradually increase the channels from basic CNN(50) to basic CNN(400). As the channel number increases, dynamics of the normalized training margins in the first row change from a phase transition with a cross-over in large margins to a monotone improvement of margin distributions. This phenomenon is not a surprise since with a strong representation power, the whole training margin distribution can be monotonically improved without sacrificing the large margins. On the other hand, the generalization or test error can never be monotonically improved. In the second row, heatmaps depict rank correlations of dynamics between training and test margin errors, which clearly show the phase transitions for CNN(50) and CNN(100) and its disappearance for CNN(400).

2. In the second experiment shown in 5, we compare the normalized margin dynamics of training CNN(400) and ResNet18 on two different datasets, CIFAR100 (the simpler) and Mini-ImageNet (the more complex). It shows that: (a) CNN(400) (5.8M parameters) does not have an over-representation power on CIFAR100, whose normalized training margin

---

[1]The Spearman's $\rho$ rank correlation measures how two variables are correlated up to a monotone transform and a larger correlation means a closer tendency.

dynamics exhibits a phase transition – a sacrifice of large margins to improve small margins during training; (b) ResNet18 (11M parameters) exhibits an over-representation power on CIFAR100 via a monotone improvement on training margins, but loses such a power in Mini-ImageNet with the phase transitions in margin dynamics.

More experiments including AlexNet and VGG16 are shown in Appendix Figure 11.

This phenomenon is not unfamiliar to us, since Breiman (Breiman, 1999) has pointed out that the improvement of training margins is not enough to guarantee a small generalization or test error in the boosting type algorithms. In this paper Breiman designed an algorithm, called *arc-gv*, enjoying an uniformly better training margin distribution comparing with Adaboost but suffer a higher generalization error. Now again we find the same phenomenon ubiquitous in deep neural networks.

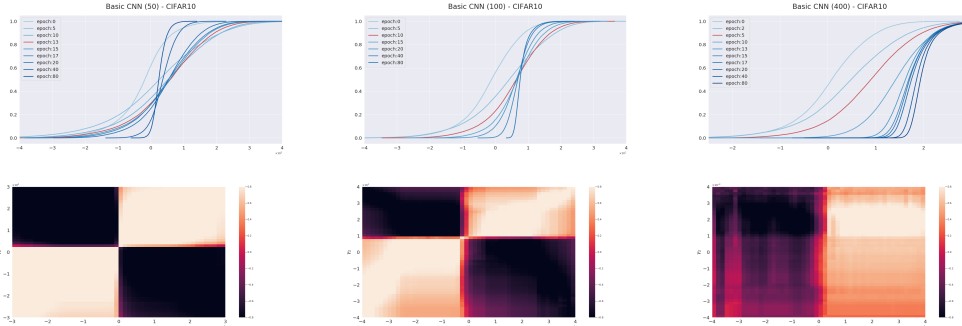

Figure 4: Breiman's Dilemma I. Net structure: Basic CNN(50) (Left), Basic CNN(100) (Middle), Basic CNN(400) (Right) . Dataset: CIFAR10 with 10 percent labels corrupted. Top: dynamics of training margin distributions. Bottom: heatmaps of Spearman's $\rho$ correlation between test margin error $\mathbb{P}[e_{\gamma_1}(\tilde{f}(x), y)]$ and training margin error $\mathbb{P}_n[e_{\gamma_2}(\tilde{f}(x_i), y_i)]$, where $(x, y)$-coordinates stand for $(\gamma_1, \gamma_2)$. With a fixed dataset, we explore how the expression power of the network influences the phase transitions of margin dynamics. The cross-over in the dynamics of training margin distributions becomes obscure and eventually disappears as the channel number increases. A clear phase transition is illustrated via the heatmap, where the training margin dynamics are highly correlated with test margin dynamics when we use Basic CNN(50) and CNN(100) (the area on the diagonal is light in the left and middle) and the training margin dynamics is very distinct to test error ($\gamma_1 \leq 0$) in CNN(100) (right).

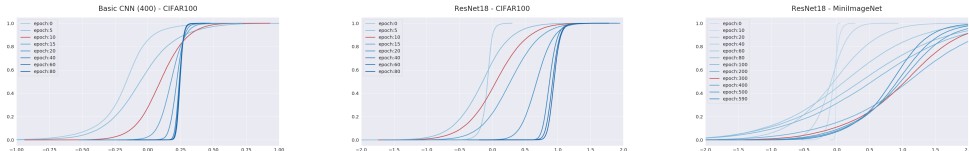

Figure 5: Breiman's Dilemma II. Net structure: Basic CNN(400) (Left), ResNet18 (Middle, Right). Dataset: CIFAR100 (Left, Middle), Mini-ImageNet (Right) with 10 percent labels corrupted. With a fixed network structure, we further explore how the complexity of dataset influences the margin dynamics. Taking ResNet18 as an example, margin dynamics on CIFAR100 doesn't have any cross-over (phase transition), but on Mini-Imagenet a cross-over occurs.

In the end, it's worth mentioning different choices of the normalization factor estimates may affect the range of predictability. In all experiments above, normalization factor is estimated via an upper bound on spectral norm given in Appendix A (Lemma A.1 in Section A). One could also use power iteration (Miyato et al., 2018) to present a more precise estimation on spectral norm. It turns out a more accurate estimation of spectral norm can extend the range of predictability, but Breiman's dilemma is still there when the balance between model expression power and dataset complexity is broken. More experiments on this aspect can be found in Figure 10 in Appendix.

## 4 CONCLUSION

In this paper, we show that Breiman's dilemma is ubiquitous in deep learning, in addition to previous studies on Boosting algorithms. We exhibit that Breiman's dilemma is closely related to the trade-off between model expression power and data complexity. A novel perspective on phase transitions in dynamics of Lipschitz-normalized margin distributions is proposed to inspect when the model has over-representation power compared to the dataset, instead of merely counting the number of parameters. A data-driven early stopping rule by monitoring the margin dynamics is a future direction to explore. Lipschitz semi-norm plays an important role in normalizing or regularizing neural networks, e.g. in GANs (Kodali et al., 2017; Miyato et al., 2018), therefore a more careful treatment deserves further pursuits.

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

## A ESTIMATE OF NORMALIZATION FACTORS

In this section we discuss how to estimate the Lipschitz constant bound in (2). Given an operator $W$ associated with a convolutional kernel $w$, i.e. $Wx = w * x$, there are two ways to estimate its operator norm. We begin with a useful lemma,

**Lemma A.1.** *For convolution operator with kernel $w$, i.e. $Wx := w * x$, there holds*

$$\|w * x\|_2 \le \|w\|_1 \|x\|_2.$$

*In other words, $\|W\|_\sigma \le \|w\|_1$.*

*Proof.*

$$
\begin{aligned}
\|w * x\|_2^2 &= \sum_u (\sum_v x(u)w(u-v))^2 \\
&= \sum_u (\sum_v (x(u)\sqrt{w(u-v)}) \cdot \sqrt{w(u-v)})^2 \\
&\le \sum_u (\sum_v x(u)^2 w(u-v))(\sum_v w(u-v)), \\
&= \|w\|_1^2 \|x\|_2^2
\end{aligned}
$$

where the second last step is due to Cauchy-Schwartz inequality. $\square$

A. $\ell_1$-norm. The convolutional operator (spectral) norm can be upper bounded by the $\ell_1$-norm of its kernels, i.e. $\|W\|_\sigma \le \|w\|_1$. This is a simple way but the bound gets loose when the channel numbers increase.

B. Power iteration. A fast approximation for the spectral norm of the operator matrix is given in (Miyato et al., 2018) in GANs that is based on power iterations (Golub & Van der Vorst, 2001). Yet as a shortcoming, it is not easy to apply to the ResNets.

We compare two estimation in Appendix Figure 10. It turns out both of them have prediction power on the tendency of generalization error and both of them will fail when the network has large enough expression power. Though using $\ell_1$ norm of kernel is extremely efficient, the power iteration method may be tighter and has a wider range of predictability.

In the remaining of this section, we will particularly discuss the treatment of ResNets. ResNet is usually a composition of the basic blocks shown in Figure 6 with short-cut structure. The following method is used in this paper to estimate the upper bound of operator or spectral norm of such a basic block of ResNet.

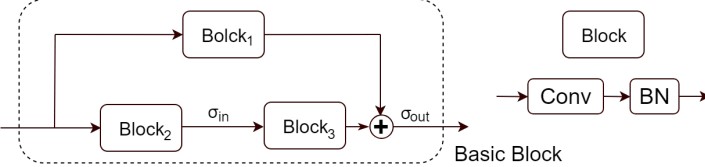

Figure 6: A basic block in ResNets used in this paper. The shortcut consists of one block with convolutional and batch-normalization layers, while the main stream has two blocks. ResNets are constructed as a cascading of several basic blocks of various sizes.

(a) Convolution layer: its operator norm can be bounded either by the $\ell_1$ norm of kernel or by power iteration above.

(b) Batch Normalization (BN): in training process, BN normalizes samples by $x^+ = (x - \mu_B)/\sqrt{\sigma_B^2 + \epsilon}$, where $\mu_B, \sigma_B^2$ are mean and variance of batch samples, while keeping an online averaging as $\hat{\mu}$ and $\hat{\sigma}^2$. Then BN rescales $x^+$ by estimated parameters $\hat{\alpha}, \hat{\beta}$ and output $\hat{x} = \hat{\alpha}x^+ + \hat{\beta}$. Therefore the whole rescaling of BN on the kernel tensor $w$

of the convolution layer is $\hat{w} = w\hat{\alpha}/\sqrt{\hat{\sigma}^2 + \epsilon}$ and its corresponding rescaled operator is $\|\hat{W}\|_\sigma = \|W\|_\sigma \hat{\alpha}/\sqrt{\hat{\sigma}^2 + \epsilon}$.

(b) Activation and pooling: their Lipschitz constants $L_\sigma$ can be known a priori, e.g. $L_\sigma = 1$ for ReLU and hence can be ignored. In general, $L_\sigma$ can not be ignored if they are in the shortcut as discussed below.

(d) Shortcut: In residue net with basic block in Figure 6, one has to treat the mainstream $(\text{Block}_2, \text{Block}_3)$ and the shortcut $\text{Block}_1$ separately. Since $\|f + g\|_{\mathcal{F}} \leq \|f\|_{\mathcal{F}} + \|g\|_{\mathcal{F}}$, in this paper we take the Lipschitz upper bound by $L_{\sigma_{\text{out}}}(\|\hat{W}_1\|_\sigma + L_{\sigma_{\text{in}}}\|\hat{W}_2\|_\sigma\|\hat{W}_3\|_\sigma)$, where $\|\hat{W}_i\|_\sigma$ denotes a spectral norm estimate of BN-rescaled convolutional operator $W_i$. In particular $L_{\sigma_{\text{out}}}$ can be ignored since all paths are normalized by the same constant while $L_{\sigma_{\text{in}}}$ can not be ignored due to its asymmetry.

## B  STRUCTURE OF BASIC CNN

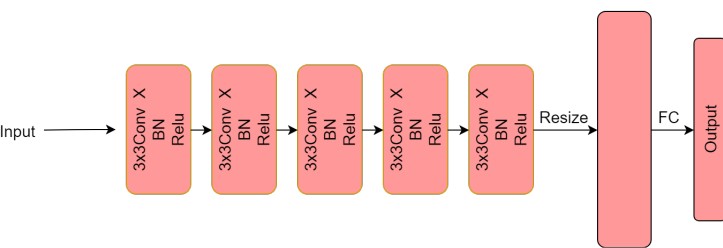

Figure 7: Illustration of the structure of basic CNN.

## C  EXPERIMENTS

### C.1  SPEARMAN'S $\rho$ RANK CORRELATION COEFFICIENT

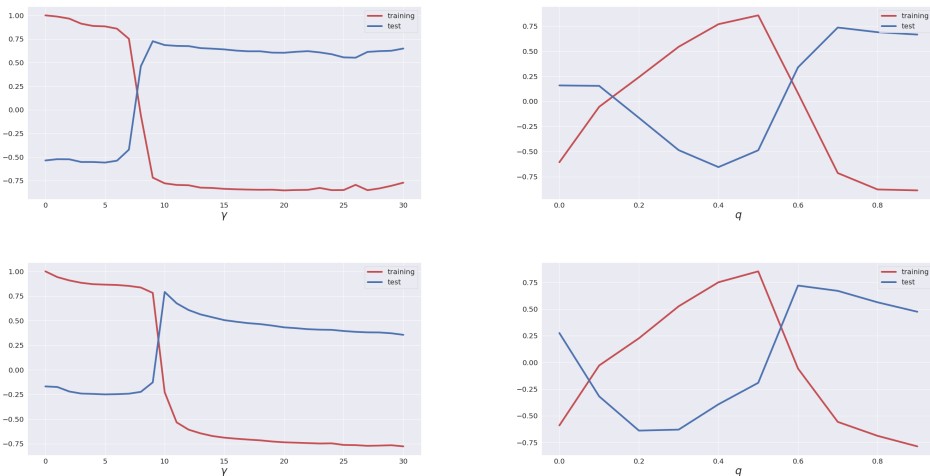

Figure 8: Spearman's $\rho$ rank correlation at different $\gamma$ and $q$. Dataset: CIFAR10 (Left) and CIFAR10 with 10 percents label corrupted (Right). Net structure: Basic CNN(50). Left: training margin error and generalization error (Blue), training error (Red). Right: inverse quantile margin and generalization error (Blue), training error (Red). The dynamic of large margin is closely related to the generalization error.

## C.2 EXAMPLES: TWO LOCAL MINIMUMS IN RESNET18

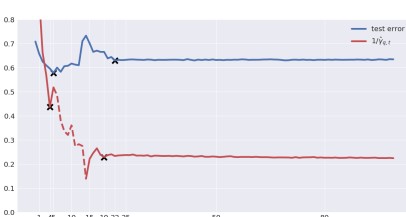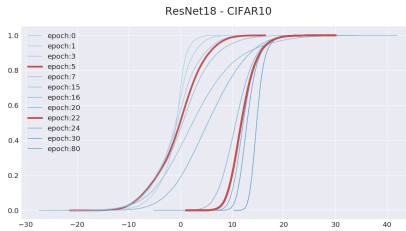

Figure 9: Dynamic of inverse quantile margin. Data, CIFAR10 with 10 percents label corrupted. Network, ResNet18. Normalization factor, spectral complexity estimated by power iteration. Left: the dynamic of generalization error and inverse quantile margin with $q = 0.95$. Overfitting occurs and two local minimums are marked with x in each dynamic. The dash line highlight where the margin distribution is uniformly improved. Right: dynamic of training margin distribution. Two distributions when local minimum of generalization error occurs are highlighted with red color. The picture is slight different here, since after the first (better) local minimum, the training margin distribution is uniformly improved without reducing generalization error. Therefore, we could not expect the inverse quantile margin to reflect the tendency of generalization error globally, especially the order of two local minimums. However, around epochs when local minimum occurs, the training margin distribution still has a cross-over, and thus the inverse quantile margin could reflect the tendency locally.

## C.3 ESTIMATED BY POWER ITERATION AND KERNEL $l_1$ NORM

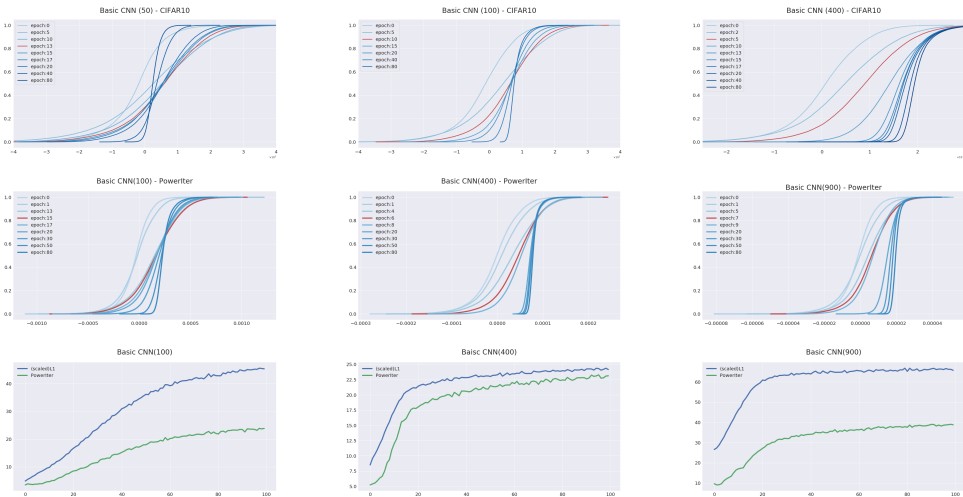

Figure 10: Power iteration: success and failure. Top: spectral norm in $L_f$ is estimated by the corresponding kernel $l_1$ norm. Bottom: spectral norm is estimated by Power Iteration. Net structure, Basic CNN with channels 50(Top, Left), 100(Top, Middle), 400(Top Right), 200(Bottom, Left), 600(Bottom, Middle), 900(Bottom, Right). Dataset: CIFAR10 with 10 percents corrupted. A more accurate estimation of spectral norm can extend the range of predictability, but eventually face Breiman's dilemma if the balance between model expression power and dataset complexity is broken.

### C.4 SUCCESS AND FAILURE IN MORE PRACTICAL NETWORK AND DATASET

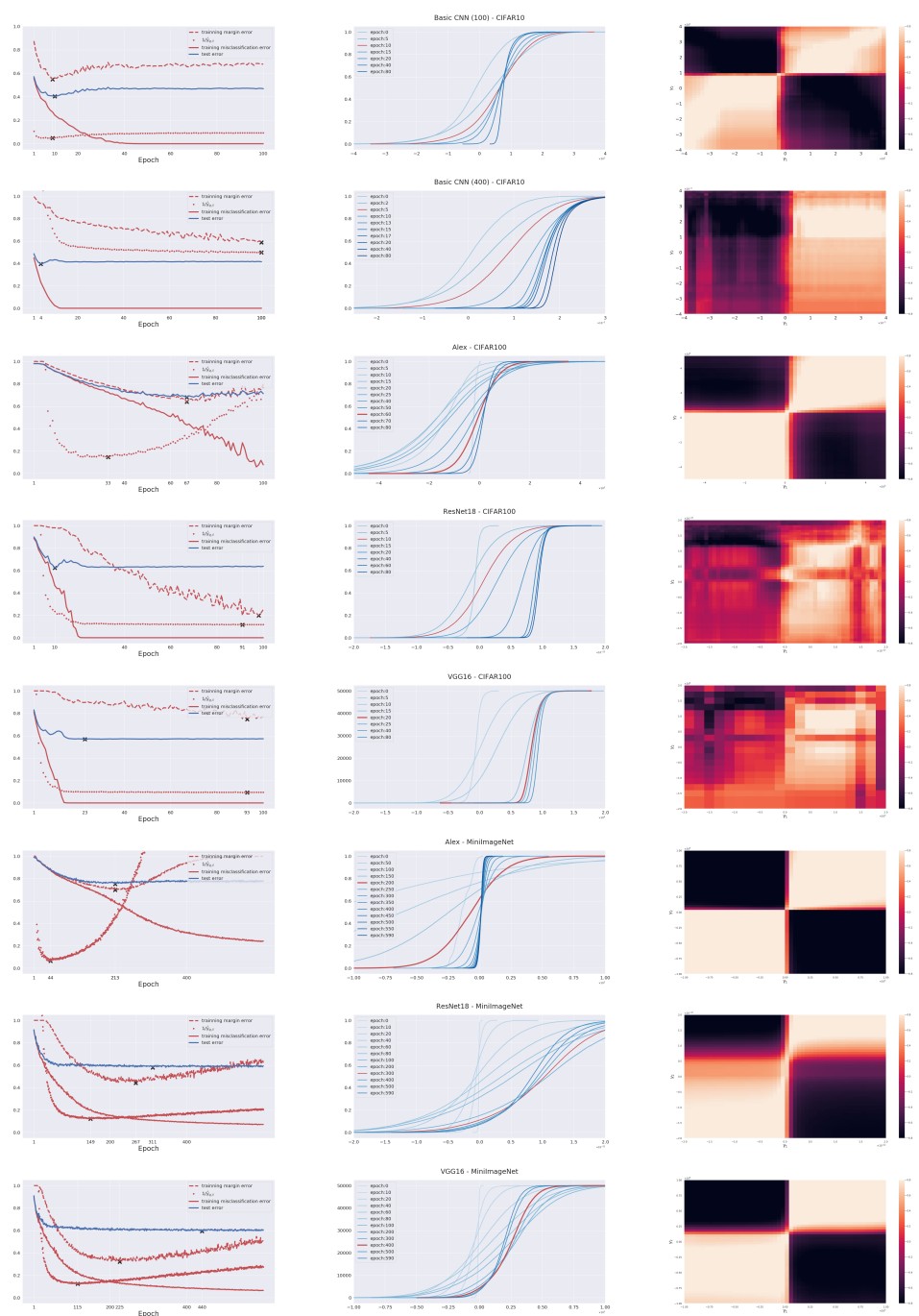

Figure 11: Examples on more practical network and dataset. The dataset and network we applied is listed in each row. Left: curve of training error, generalization error, training margin error and inverse quantile margin. Middle: dynamic of training margin distribution. Right: heatmap of Spearman's $\rho$ correlation between test margin error $\mathbb{E}[e_{\gamma_1}(\tilde{f}(x), y)]$ and training margin error $(1/n) \sum [e_{\gamma_2}(\tilde{f}(x_i), y_i)]$ against $(\gamma_1, \gamma_2)$.

## C.5 DYNAMIC OF TEST MARGIN DISTRIBUTION

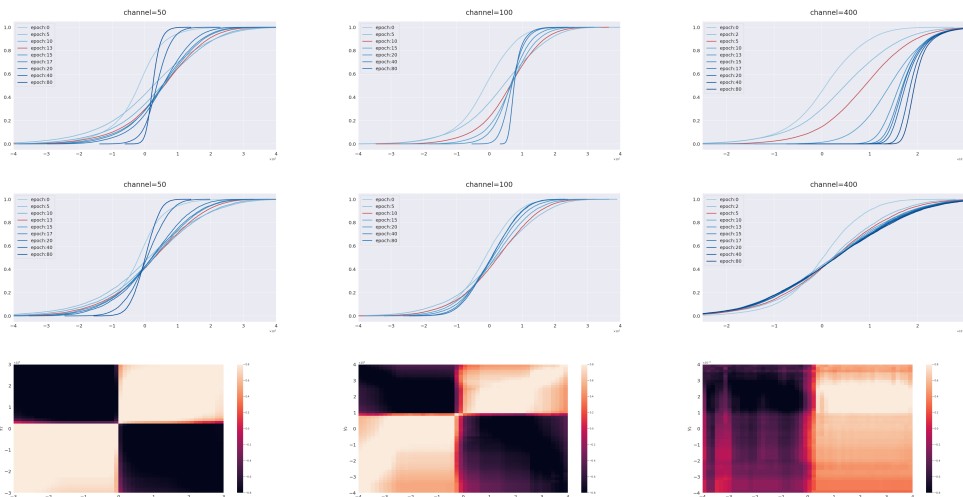

Figure 12: Comparison between dynamic of test margin distribution and training margin distribution. Top: training margin distribution. Bottom: test margin distribution. Net structure: Basic CNN with channels 50 (Left), 100 (Middle) and 400 (Right). When model becomes complex, the dynamic of training margin distribution lose the predictability on test margin distribution.

## D PROOFS

### D.1 AUXILIARY LEMMAS

Lemma 2.1 follows from the following Lemma D.1 by applying it to bounded function $\ell_\gamma(f(x), y)$ whose range is $[0, 1]$.

**Lemma D.1.** *For any $\delta \in (0, 1)$ and bounded-value functions $\mathcal{F}_B := \{f : \mathcal{X} \to \mathbb{R} : \|f\|_\infty \leq B\}$, the following holds with probability at least $1 - \delta$,*

$$\sup_{f \in \mathcal{F}_B} \mathbb{E}_n f(x) - \mathbb{E} f(x) \leq 2\mathcal{R}_n(\mathcal{F}_B) + B\sqrt{\frac{\log(1/\delta)}{2n}} \tag{13}$$

*where*

$$\mathcal{R}_n(\mathcal{F}) = \mathbb{E} \sup_{f \in \mathcal{F}} \frac{1}{n} \sum_{i=1}^{n} \varepsilon_i f(x_i) \tag{14}$$

*is the Rademacher Complexity of function class $\mathcal{F}$.*

For completeness, we include its proof that also needs the following well-known McDiarmid's inequality (see, e.g. Wainright (2019)).

**Lemma D.2** (McDiarmid's Bounded Difference Inequality). *For $B_i$-bounded difference functions $h : \mathcal{X} \to \mathbb{R}$ s.t. $|h(x_i, x_{-i}) - h(x'_i, x_{-i})| \leq B_i$,*

$$\mathbb{P}\{\mathbb{E}_n h - \mathbb{E}_x h(x) \geq \varepsilon\} \leq \exp\left(-\frac{2\epsilon^2}{\sum_{i=1}^{n} B_i^2}\right),$$

*Proof of Lemma D.1.* It suffices to show that for $\bar{f} = f(x) - \mathbb{E}f(x)$,

$$\sup_{f \in \mathcal{F}_B} \mathbb{E}_n \bar{f} = \sup_{f \in \mathcal{F}_B} \mathbb{E}_n \bar{f} - \mathbb{E} \sup_{f \in \mathcal{F}_B} \mathbb{E}_n \bar{f} + \mathbb{E} \sup_{f \in \mathcal{F}_B} \mathbb{E}_n \bar{f} \tag{15}$$

where with probability at least $1 - \delta$,

$$\sup_{f \in \mathcal{F}_B} \mathbb{E}_n \bar{f} - \mathbb{E} \sup_{f \in \mathcal{F}_B} \mathbb{E}_n \bar{f} \leq B \sqrt{\frac{\log 1/\delta}{2n}} \tag{16}$$

by McDiarmid's bounded difference inequality, and

$$\mathbb{E} \sup_{f \in \mathcal{F}_B} \mathbb{E}_n \bar{f} \leq 2\mathcal{R}_n(\mathcal{F}) \tag{17}$$

using Rademacher complexity.

To see (16), we are going to show that $\sup_{f \in \mathcal{F}_B} \mathbb{E}_n \bar{f}$ is a bounded difference function. Consider $g(x_1^n) = \mathbb{E}_n \bar{f} = \frac{1}{n} \sum_{i=1}^n f(x_i) - \mathbb{E}_x f(x)$. Assume that the $i$-th argument $x_i$ changes to $x_i'$, then for every $g$,

$$
\begin{aligned}
g(x_i, x_{-i}) - \sup_g g(x_i', x_{-i}) &\leq& g(x_i, x_{-i}) - g(x_i', x_{-i}) \\
&\leq& \frac{1}{n}[f(x_i) - f(x_i')] \\
&\leq& \frac{B}{n}.
\end{aligned}
$$

Hence $\sup_g g(x_i, x_{-i}) - \sup_g g(x_i', x_{-i}) \leq B/n$, which implies that $\sup_{f \in \mathcal{F}_B} \mathbb{E}_n \bar{f}$ is a $B/n$-bounded difference function. Then (16) follows from the McDiarmid's inequality (Lemma D.2) using $B_i = B/n$ and $\delta = \exp(-2n\varepsilon^2/B^2)$.

As to (17),

$$
\begin{aligned}
\mathbb{E} \sup_{f \in \mathcal{F}_B} \mathbb{E}_n \bar{f} &=& \mathbb{E}_{x_1^n} \sup_{f \in \mathcal{F}_B} \mathbb{E}_{y_1^n} \left[ \mathbb{E}_n f(x_1^n) - \mathbb{E}_n f(y_1^n) \right] \\
&\leq& \mathbb{E}_{x_1^n, y_1^n} \sup_{f \in \mathcal{F}_B} \left[ \mathbb{E}_n f(x_1^n) - \mathbb{E}_n f(y_1^n) \right] \\
&=& \mathbb{E}_{x_1^n, y_1^n} \sup_{f \in \mathcal{F}_B} \mathbb{E}_{\varepsilon_1^n} \frac{1}{n} \sum_{i=1}^n \varepsilon_i \left( f(x_i) - f(y_i) \right), \ \ \varepsilon_i \in \{\pm 1\} \sim \mathcal{B}(n, 1/2) \\
&\leq& \mathbb{E}_{x_1^n, y_1^n, \varepsilon_1^n} \sup_{f \in \mathcal{F}_B} \frac{1}{n} \sum_{i=1}^n (\varepsilon_i f(x_i) - \varepsilon_i f(y_i)) \\
&\leq& 2 \mathbb{E}_{x_1^n, \varepsilon_1^n} \sup_{f \in \mathcal{F}_B} \frac{1}{n} \sum_{i=1}^n \varepsilon_i f(x_i) = 2\mathcal{R}(\mathcal{F}_B)
\end{aligned}
$$

that ends the proof. $\square$

We also need the following contraction inequality of Rademacher Complexity (Ledoux & Talagrand, 1991; Meir & Zhang, 2003).

**Lemma D.3** (Rademacher Contraction Inequality). *For any Lipschitz function: $\phi : \mathbb{R} \to \mathbb{R}$ such that $|\phi(x) - \phi(y)| \leq L|x - y|$,*

$$\mathcal{R}(\phi \circ \mathcal{F}) \leq L\mathcal{R}(\mathcal{F}).$$

Ledoux & Talagrand (1991) has an additional factor 2 in the contraction inequality which is dropped in Meir & Zhang (2003). Its current form is stated in Mohri et al. (2012) as Talagrand's Lemma (Lemma 4.2).

Beyond, we further introduce the family,

$$\mathcal{G} = \{g(x, y) = \zeta(f(x), y) : \mathcal{X} \times \mathcal{Y} \to \mathbb{R}, f \in \mathcal{F}\}, \tag{18}$$

and the sub-family constraint in Lipschitz semi-norm on $f$,

$$\mathcal{G}_L = \{g(x, y) = \zeta(f(x), y) : \mathcal{X} \times \mathcal{Y} \to \mathbb{R}, f \in \mathcal{F} \text{ with } \|f\|_{\mathcal{F}} \leq L\}. \tag{19}$$

The following lemma (Koltchinskii et al., 2002) allows us to bound the Rademacher complexity term of $\mathcal{R}_n(\mathcal{G})$ by $\mathcal{R}_n(\mathcal{H})$,

**Lemma D.4.** $\mathcal{R}_n(\mathcal{G}_L) \leq K^2 \mathcal{R}_n(\mathcal{H}_L)$

*Proof of Lemma D.4.*

$$
\begin{aligned}
\mathcal{R}_n(\mathcal{G}_L) &= \frac{1}{n}\mathbb{E}_{S,\epsilon} \sup_{\|f\|\leq L} \sum_{i=1}^{n} \epsilon_i \zeta(f(x_i), y_i), \\
&= \frac{1}{n}\mathbb{E}_{S,\epsilon} \sup_{\|f\|\leq L} \sum_{i=1}^{n} \sum_{y\in\mathcal{Y}} \epsilon_i \zeta(f(x_i), y_i) 1[y_i = y], \\
&= \frac{1}{n} \sum_{y\in\mathcal{Y}} \mathbb{E}_{S,\epsilon} \sup_{\|f\|\leq L} \sum_{i=1}^{n} \epsilon_i \zeta(f(x_i), y_i) 1[y_i = y], \\
&= \frac{1}{n} \sum_{y\in\mathcal{Y}} \mathbb{E}_{S,\epsilon} [\sup_{\|f\|\leq L} \sum_{i=1}^{n} \epsilon_i (\frac{2\cdot 1[y_i = y] - 1}{2} + \frac{1}{2})], \\
&\leq \frac{1}{2n} \sum_{y\in\mathcal{Y}} \mathbb{E}_{S,\epsilon} [\sup_{\|f\|\leq L} \sum_{i=1}^{n} \epsilon_i (\frac{2\cdot 1[y_i = y] - 1)}{2}] + \frac{1}{2n} \sum_{y\in\mathcal{Y}} \mathbb{E}_{S,\epsilon} [\sup_{\|f\|\leq L} \sum_{i=1}^{m} \epsilon_i \zeta(f(x_i), y)]), \\
&= \frac{1}{n} \sum_{y\in\mathcal{Y}} \mathbb{E}_{S,\epsilon} [\sup_{\|f\|\leq L} \sum_{i=1}^{m} \epsilon_i \zeta(f(x_i), y)]), \\
&= \frac{1}{n} \sum_{y\in\mathcal{Y}} \mathbb{E}_{S,\epsilon} [\sup_{\|f\|\leq L} \sum_{i=1}^{m} \epsilon_i ([f(x_i)]_y - \max_{y'\neq y}[f(x_i)]_{y'})], \\
&\leq \frac{1}{n} \sum_{y\in\mathcal{Y}} \mathbb{E}_{S,\epsilon} [\sup_{\|f\|\leq L} \sum_{i=1}^{m} \epsilon_i [f(x_i)]_y] + \frac{1}{n} \sum_{y\in\mathcal{Y}} \mathbb{E}_{S,\epsilon} [\sup_{\|f\|\leq L} \sum_{i=1}^{m} \epsilon_i \max_{y'\neq y}[f(x_i)]_{y'}], \\
&\leq \frac{K}{n} \mathbb{E}_{S,\epsilon} [\sup_{h\in\mathcal{H}_L} \sum_{i=1}^{m} \epsilon_i h(x_i)] + \frac{K(K-1)}{n} \mathbb{E}_{S,\epsilon} [\sup_{h\in\mathcal{H}_L} \sum_{i=1}^{m} \epsilon_i h(x_i)], \\
&= K^2 \mathcal{R}_n(\mathcal{H}_L),
\end{aligned}
$$

where the last inequality is implied from $\mathcal{R}_n(\{\max(f_1,\ldots,f_M) : f_i \in \mathcal{F}_i\}) \leq \sum_{m=1}^{M} \mathcal{R}_n(\mathcal{F}_m)$ (Koltchinskii et al., 2002; Mohri et al., 2012). □

### D.2 PROOF OF PROPOSITION 1

*Proof of Proposition 1.* Without loss of generality, we assume $L_{\sigma_i} = 1, i = 1, \ldots, l$. Let $\mathcal{T}(r) =: \{t(x) = w \cdot x : \|w\|_2 \leq r\}$ be the class of linear function with Lipschitz semi-norm less than $r$ and we show that for each $t \in \mathcal{T}(L/2)$, there exists $f \in \mathcal{F}$ with $\|f\|_{\mathcal{F}} \leq L$ and $y_0 \in \{1, \ldots, K\}$ such that $h(x) = [f(x)]_{y_0} \in \mathcal{H}_L$. Let's $t(x) = w_0 \cdot x$ with $\|w_0\|_2 \leq L/2$, we construct the network $f(x) = W_l \sigma_l(x_l) + b_l, x_i = \sigma_i(W_{i-1}x_{i-1} + b_{i-1}), i = 1, \ldots, l, x_0 = x$ as follows,
- $x_1 = \sigma_1(W_1 x) = (\sigma_1(w_0 \cdot x), \sigma_1(-w_0 \cdot x), 0, \ldots, 0)$
- $x_k = \sigma_k(W_k x_{k-1}) = (\sigma_k([x_{k-1}]_1), \sigma_k([x_{k-1}]_2), 0, \ldots, 0), k = 2, \ldots, l-1$
- $x_l = W_l x_{l-1} = ([x_{l-1}]_1 - [x_{l-1}]_2, 0, \ldots, 0)$
By construction above, we let $h(x) = [f(x)]_1$,

$$
\begin{aligned}
h(x) &= \sigma_1(w_0 \cdot x) - \sigma_1(-w_0 \cdot x), \\
&= w_0 \cdot x,
\end{aligned}
$$

where $\|f\|_{\mathcal{F}} \le \Pi_{i=1}^l \|W_i\|_\sigma = 2L/2 = L$, and thus $h \in \mathcal{H}_L$ by definition. Therefore,

$$\mathcal{R}_n(\mathcal{H}_L) \ge \mathcal{R}_n(\mathcal{T}(L/2)),$$

$$= \mathbb{E}_S \mathbb{E}_\epsilon \sup_{\|w\| \le L/2} \sum_{i=1}^n \epsilon_i w \cdot x_i,$$

$$= \frac{L}{2} \mathbb{E}_S \mathbb{E}_\epsilon \| \sum_{i=1}^n \epsilon_i x_i \|_2,$$

$$\ge CL\mathbb{E}_S \sqrt{\sum_{i=1}^n \|x_i\|_2},$$

where the second equality is implied from Cauchy-Schwarz inequality and the last inequality is implied from Khintchine inequality.

$\square$

## D.3 PROOF OF THEOREM 1

*Proof of Theorem 1.* Consider $l_{(\gamma_1,\gamma_2)}(\zeta(\tilde{f}(x), y))$, where $\tilde{f} := f/L_f$ is the normalized network, $\zeta(\tilde{f}(x), y) \in \mathcal{G}_1$. Then for any $\gamma_2 > \gamma_1 \ge 0$,

$$P[\zeta(\tilde{f}(x), y) < \gamma_1] \le P[\ell_{(\gamma_1,\gamma_2)}(\zeta(\tilde{f}(x), y)],$$

$$\le \mathbb{P}_n \ell_{(\gamma_1,\gamma_2)}(\tilde{f}(x), y) + 2\mathcal{R}_n(l_{(\gamma_1,\gamma_2)} \circ \mathcal{G}_1) + \sqrt{\frac{\log(1/\delta)}{2n}},$$

$$\le \mathbb{P}_n \ell_{(\gamma_1,\gamma_2)}(\tilde{f}(x), y) + \frac{2}{\Delta}\mathcal{R}_n(\mathcal{G}_1) + \sqrt{\frac{\log(1/\delta)}{2n}},$$

$$\le \mathbb{P}_n \ell_{\gamma_1,\gamma_2}(\tilde{f}(x), y) + \frac{2K^2}{\Delta}\mathcal{R}_n(\mathcal{H}_1) + \sqrt{\frac{\log(1/\delta)}{2n}},$$

$$\le \mathbb{P}_n \ell_{\gamma_2}(\tilde{f}(x), y) + \frac{2K^2}{\Delta}\mathcal{R}_n(\mathcal{H}_1) + \sqrt{\frac{\log(1/\delta)}{2n}},$$

where the first and last inequality is implied from $1[\zeta < \gamma_1] \le \ell_{(\gamma_1,\gamma_2)}(\zeta) \le 1[\zeta < \gamma_2]$, the second inequality is a direct consequence of Lemma D.1, the third inequality results from Rademacher Contraction Inequality (Lemma D.3) and finally the fourth equation is implied from Lemma D.4.

$\square$

## D.4 PROOF OF THEOREM 2

*Proof of Theorem 2.* Firstly, we show after normalization, the normalize margin has an upper bound,

$$\|f(x)\|_2 = \|\sigma_L(W_L x_{L-1} + b_L)\|_2,$$

$$\le L_{\sigma_L} \|W_L x_{L-1} + b_L\|_2,$$

$$\le (L_{\sigma_L} \|\bar{W}_L\|_\sigma)(\|x_{L-1}\|_2 + 1)$$

$$\cdots$$

$$\le \Pi_{i=1}^L (L_{\sigma_i} \|\bar{W}_i\|_\sigma) \|x\|_2 + \Sigma_{i=1}^L (\Pi_{j=i}^L (L_{\sigma_i} \|\bar{W}_i\|_\sigma)),$$

where $x_i = \sigma_i(W_i x_{i-1} + b_i)$ with $x_0 = x$, $\bar{W}_i = (W_i, b_i)$ and $L_{\sigma_i}$ is the Lipschitz constant of activation function $\sigma_i$ with $\sigma_i(0) = 0, i = 1, \ldots, L$. Then, for normalized network $\tilde{f} = f/L_f$ with

$L_f = \Pi_{i=1}^L (L_{\sigma_i} \|\bar{W}_i\|_\sigma)$ and $\|x\|_2 \leq M$,

$$\|\tilde{f}(x)\|_2 \leq M + L.$$

Therefore $\zeta(\tilde{f}(x), y) \leq 2\|\tilde{f}(x)\|_2 = 2(M + L) =: M_1$, and the quantile margin is also bounded $\hat{\gamma}_{q,t} \leq M_1$ for all $q \in (0, 1), t = 1, \ldots, T$.

The remaining proof is standard. For any $\epsilon > 0$, we take a sequence of $\epsilon_k$ and $\gamma_k, k = 1, 2, \ldots$ by $\epsilon_k = \epsilon + \sqrt{\frac{\log k}{n}}$ and $\gamma_k = M_1 2^{-k}$. Then by Theorem 1,

$$\mathbb{P}(A_k) \leq \exp(-2n\epsilon_k^2),$$

where $A_k$ is the event $\mathbb{P}[\zeta(\tilde{f}_t(x), y) < 0] > \mathbb{P}_n[\zeta(\tilde{f}(x), y) < \gamma_k] + \frac{2K^2}{\gamma_k}\mathcal{R}(\mathcal{H}_1) + \epsilon_k$, and the probability is taken over samples $\{x_1, \ldots x_n\}$. We further consider the probability for none of $A_k$ occurs,

$$\mathbb{P}(\exists A_k) \leq \Sigma_{k=1}^\infty P(A_k),$$

$$\leq \Sigma_{k=1}^\infty \frac{1}{k^2} \exp(-2n\epsilon^2),$$

$$\leq 2 \exp(-2n\epsilon^2).$$

Hence, fix a $q \in [0, 1]$, for any $t = 1, \ldots, T$, as long as $\hat{\gamma}_{q,t} > 0$, there exists a $\hat{k} \geq 1$ such that,

$$\gamma_{\hat{k}+1} \leq \hat{\gamma}_{q,t} < \gamma_{\hat{k}}. \tag{20}$$

Therefore,

$$A_{\hat{k}+1} \supseteq \mathbb{P}[\zeta(\tilde{f}_t(x), y) < 0] > \mathbb{P}_n[\zeta(\tilde{f}_t(x), y) < \hat{\gamma}_{q,t}] + \frac{2K^2}{\gamma_{\hat{k}+1}}\mathcal{R}(\mathcal{H}_1) + \epsilon_{\hat{k}+1},$$

$$\supseteq \mathbb{P}[\zeta(\tilde{f}_t(x), y) < 0] > \mathbb{P}_n[\zeta(\tilde{f}_t(x), y) < \hat{\gamma}_{q,t}] + \frac{4K^2}{\hat{\gamma}_{q,t}}\mathcal{R}(\mathcal{H}_1) + \epsilon_{\hat{k}+1},$$

$$= \mathbb{P}[\zeta(\tilde{f}_t(x), y) < 0] > \mathbb{P}_n[\zeta(\tilde{f}_t(x), y) > \hat{\gamma}_{q,t}] + \frac{4K^2}{\hat{\gamma}_{q,t}}\mathcal{R}(\mathcal{H}_1) + \epsilon + \sqrt{\frac{\log(\hat{k}+1)}{n}},$$

$$\supseteq \mathbb{P}[\zeta(\tilde{f}_t(x), y) < 0] > \mathbb{P}_n[\zeta(\tilde{f}_t(x), y) > \hat{\gamma}_{q,t}] + \frac{4K^2}{\hat{\gamma}_{q,t}}\mathcal{R}(\mathcal{H}_1) + \epsilon + \sqrt{\frac{\log\log_2(2M_1/\hat{\gamma}_{q,t})}{n}}.$$

The first inequality is implied from $\mathbb{P}_n[\zeta(\tilde{f}_t(x), y) < \hat{\gamma}_{q,t}] > \mathbb{P}_n[\zeta(\tilde{f}_t(x), y) < \gamma_{\hat{k}+1}]$, since $\gamma_{\hat{k}+1} \leq \hat{\gamma}_{q,t}$. The second inequality is implied from $\hat{\gamma}_{q,t} < 2\gamma_{\hat{k}+1}$ and thus, $1/\gamma_{\hat{k}+1} < 2/\hat{\gamma}_{q,t}$. The third equality is the direct definition of $\epsilon_{\hat{k}}$. The last inequality is implied from $\hat{k} + 1 = \log_2(M_1/\gamma_{\hat{k}+1})$ and again, $1/\gamma_{\hat{k}+1} < 2/\hat{\gamma}_{q,t}$. The conclusion is proved immediately if we do a transform from $\epsilon$ to $\delta$. $\square$

