# OpenReview forum: "ON BREIMAN’S DILEMMA IN NEURAL NETWORKS: SUCCESS AND FAILURE OF NORMALIZED MARGINS"
_ICLR.cc/2019/Conference_

### Official Review · AnonReviewer1 · 2018-10-29
**.**

**Rating:** 5
**Confidence:** 3

**Review:**

The submission explores Breiman's dilemma: training margin is not always a good predictor of test error.

In particular, the authors show that:

- For under-parametrized CNNs, the training prediction margin is a good predictor of the test error.
- For over-parametrized CNNs, the training prediction margin is not a good predictor of the test error.

Throughout the submission, I suspect that the authors compute the "functional margin", that is, the difference between the largest label score and the second largest score, for correctly classified examples. Functional margins ignore the smoothness of the underlying function, a critical factor for generalization. For instance, the function f(x) = 1[x > 0] has large functional margin, but any perturbation around the x-origin would drastically change the prediction. For this reason, I think the authors should consider the "geometrical margin" instead, which is unfortunately difficult to compute for general neural networks. Their theory tries to reflect on this issue by using spectrally-normalized bounds, but the practice ignores this issue completely (as far as I can tell).

Therefore, we may be looking at the wrong statistic to predict generalization error. Is Breiman dilemma solved by re-defining margin properly? Geometrical margin can be computed in closed-form for linear classifiers, so perhaps this would be a first step in this investigation.

---

> ### Author Response · Authors · 2018-11-21
> **Response to AnonReviewer1**
>
> We thank you for your comments. We give a response to your concerns below.
>
> -- Geometric margin
> Thanks for the suggestion on geometric margin, whose definition is currently not available for deep neural networks. On the other hand, functional margins and networks Lipschitz via spectral norm products have been recently used to control the network complexity (Bartlett et al. 2017). Our results show that only a control on spectral norm and the functional margin is not sufficient to prevent the Breiman’s dilemma. Other weight norms like 2-1 norm might be worth to explore as our next direction.
>
> -- Is Breiman’s dilemma solved by re-defining margin properly?
> This is a good question. Although we don’t have a geometric margin in hand as in linear networks yet, different normalizations on functional margins may affect when Breiman’s dilemma appears as model complexity increases. On the other hand, we have seen that Breiman’s dilemma is ubiquitous in neural networks as the expressive power grows as in boosting, and what we lack is a proper complexity control at this moment that needs to be explored in the future.

---

### Official Review · AnonReviewer3 · 2018-11-02
**This paper started from Breiman's dilemma and showed that it relies on dynamics of normalized margin distribution.**

**Rating:** 5
**Confidence:** 4

**Review:**

The authors found that general generalization bounds fail to capture the ramp loss. However, once the network scaled by its Lipschitz constant, it becomes efficient to get an upper bound of generalization error, while also needs to trade-off the constant in the margin error. Due to the limitation of fixing the constant in margin error, the authors  tried to use the quantile margin to change the bound, which is easy to tune the hyper-parameter. They also conducted the experiments that the quantile margin generalization bound could be used to predict the tendency of loss curve both in training and test in some sense.

It's really an interesting work to provide a way for early stopping and to show the quantile margin maybe a substitution of tendency in training error as well as test error.

Questions:

It could be difficult to judge from the phase transition, if exists, in the evolution of normalized margin distributions curve. Maybe  some quantitative descriptions are needed.

Besides, the authors' quantile margin bound (Theorem 2) shows the upper bound of margin (or say margin error). But the bound is not direct to support the powerful experiments results, the relationship between the tendency of quantile margin, training and test error.

Typos:
 In Eqn. (10), the first $f_t$ should be $\widetilde{f}_t$ .
In Eqn. (9) and (11), there is  $1$.
In Proof in Lemma A.1, the convolution operator is $x(v)$ not $x(u)$, since Lemma is also true.
In Proof in Lemma D.4, though the proof is same in the book `Foundations of machine learning' by Mehryar Mohri, Afshin Rostamizadeh, and Ameet Talwalkar, please check the typo.
In Proof of Proposition 1, lack $\frac{1}{n}$ in Rademacher complexity.
In Proof of Theorem 1, maybe you should take $\mathbb{E}$ not $\mathbb{P}$ before $\ell_{\gamma_1,\gamma_2}(\xi(\widetilde{f}(x,y))))$.
In Proof of Theorem 2, page 18 the last line in the equation, why can the second term after divided by $L_f$ bounded by $L$, maybe need some conditions or I missed something.

---

> ### Author Response · Authors · 2018-11-21
> **Response to AnonReviewer3**
>
> We thank you for your comments. We give a response to your two major questions below.
>
> -- It’s difficult to judge phase transition in the evolution.
> Phase transitions of normalized margin evolutions can be judged via the rank correlations between training- and test- margin dynamics, shown in the second row (heatmaps) of Figure. When these heatmaps exhibit a block diagonal structure, as the left and middle ones, training- and test- margin dynamics share a similar phase transition; on the other hand, the left-right two-block structure in the right heatmap indicates the distinct phase transitions in training- and test- margin evolutions (training margins undergo a uniform improvements while test margins undergo an increase-decrease phase transition). Moreover, the cross-over in the training margin dynamics also marks the occurrence of phase-transition as well, as illustrated by the right two figures in Figure 1.
>
> -- The bound is not direct to support the results of the experiments.
> Unfortunately, existing Rademacher complexity based bounds are too loose to quantitatively calibrate the success case and failure case given models and datasets. They only provide a qualitative explanation such that: 1) when training margin dynamics share a similar phase transition with test margin dynamics, Rademacher complexity of normalized networks can be regarded as a constant without affecting the successful prediction of test error trend by training margins, 2) when training margin dynamics are uniformly improved as a distinct phase transition to test margin dynamics, such a prediction fails, Breiman’s dilemma happens, and Rademacher complexity of normalized networks blows up. Better complexity upper bounds with tighter explanation power are our future pursuit.

---

### Official Review · AnonReviewer2 · 2018-11-03
**The technical contribution is minor**

**Rating:** 4
**Confidence:** 4

**Review:**

Summary:
The authors investigate the Breiman’s dilemma in the context of deep learning. They show generalization bounds in terms of the margin distribution. They also perform experiments showing the Breiman’s dilemma.

Comments:
I am afraid the authors miss an important related paper:

Lev Reyzin, Robert E. Schapire:
How boosting the margin can also boost classifier complexity. ICML 2006: 753-760

Reyzin and Schapire explain the Breiman’s dilemma based on base classifiers’ complexity. In particular, their experiments show that arc-gv tends to use more complex decision trees than AdaBoost while it achieves better margin distribution over sample. That is, not only margin distribution, but also the complexity of base classifiers’ class matters. This is already explained by known Rademacher complexity based margin bounds.

As for quiantile-based analyses on margin bounds the following result is known:

Liwei Wang et al: A Refined Margin Analysis for Boosting Algorithms via Equilibrium Margin,
Journal of Machine Learning Research 12 (2011) 1835-1863.

They proved a shaper bound using the notion of equibrium margin. The authors should compare the presented results with this.

The technical results of the paper look quite similar to known margin bounds and I am afraid the contribution is minor or redundant.

After the rebuttal:
I read the authors' comments and understand more the technical results. I raised my score. But I still feel that the techniccal contribution is a bit weak.

---

> ### Author Response · Authors · 2018-11-21
> **Response to AnonReviewer2**
>
> We thank you for your comments.
>
> First of all, let’s recapture our main contribution to clarifying the distinctions to existing works in boosting that mentioned by the reviewer. For neural networks, we show that Breiman’s Dilemma in neural networks can be judged by the dynamics of normalized margin distributions and phase transitions therein:
>
> (A) When dynamics of training- and test- normalized margins share a similar phase transition of increase-decrease dynamics, one can predict the trend (e.g. early stopping regularization) of test margins/errors from training margins via two simplifications of traditional Rademacher complexity based bounds (e.g. Bartlett et al. 2017), by regarding as a small constant the Rademacher complexity of spectrally normalized networks.
> (B) On the other hand, when training normalized margins are uniformly improved which indicates the over-expressiveness of the models against data, such a prediction fails and the Rademacher complexity of spectrally normalized networks might be too large to make the bounds trivial. This is a similar observation in boosting reported by Breiman (1999) that a uniform improvement of margin distribution does not necessarily mean improvement in generalization performance.
>
> The two references you mentioned are good in explaining Breiman’s dilemma in boosting algorithms, but NOT applicable to deep neural networks that will be explained as follows.
>
> 1) Reyzin and Schapire (2006) shows that arc-gv (Breiman, 1999) has larger base classifier complexities (in terms of tree depth) than AdaBoost that explains its drop of generalization performance. This work has the same spirit as what we have observed for deep neural networks. But complexity measure of neural networks, where we turn to a simplification of recent spectral-normalization bounds by Bartlett et al. 2017, is different to that of convex combinations of voting classifiers in boosting.
> 2) Liwei Wang et al. (2011) proposes a generalization bound for boosting based on VC-dimensions and optimized quantile margins. First, such optimized quantile margins become trivial or do not work when training margin distributions are uniformly improved in (B) scenario above, even in boosting set. Second, most of the models in deep neural networks like the ones we studied in this paper (CNN.400, AlexNet, VGG16, ResNet), are overparameterized that VC-dimension is too loose to give trivial bound. Lately, since Bartlett (1997), neural network society has been exploring weight-norm based or size-independent bounds on Rademacher complexities, e.g. the bounds in Bartlett et al. (2017). In this paper, we are not aiming to provide a tighter estimate of such bounds; instead, we show that under scenario (A) above, Rademacher complexity can be simplified to be a constant which does not affect the prediction of test margins/errors via training margins, in a similar spirit of Liao et al. 2018 in the study of cross-entropy loss.
>
> Reference:
> [1] Bartlett, Peter L. "For valid generalization the size of the weights is more important than the size of the network." Advances in neural information processing systems. 1997.
> [2] Liao, Qianli, et al. "A surprising linear relationship predicts test performance in deep networks." arXiv preprint arXiv:1807.09659 (2018).
> [3] Bartlett, Peter L., Dylan J. Foster, and Matus J. Telgarsky. "Spectrally-normalized margin bounds for neural networks." Advances in Neural Information Processing Systems. 2017.

---

### Meta-Review · Area_Chair1 · 2018-12-18

**Confidence:** 4
**Recommendation:** Reject

**Metareview:**

The reviewers reached a consensus that the paper is not ready for publication in ICLR. (see more details in the reviews below. )